# Structured Shape-Patterns from a Sketch: A Multi-Scale Approach

Pauline Olivier*        Pooran Memari†        Marie-Paule Cani‡

LIX, École Polytechnique, CNRS, IP Paris

## ABSTRACT

Structured 2D patterns formed by the anisotropic distribution of arbitrary shapes are ubiquitous in nature and man-made environments. They may include both bounded and unbounded (extended fiber-like) shapes. In this work, we address the problem of interactively generating such patterns from a single exemplar sketched by a user. We build our solution on a new data structure, the *Support Structure Hierarchy*, computed from a multi-resolution analysis of the input exemplar, that encodes the main anisotropy directions at different scales as well as deviations from them. We propose an efficient method based on this structure to synthesize a similar distribution of shapes in an extended 2D domain. The user can also choose to hybridize multiple input exemplars by combining structural shapes extracted at different scales. As shown in a user study, our multi-scale solution generates structured shape-patterns that perceptually compete with state-of-the-art methods, whether learning-based or not. Moreover, our interactive solution, which requires no pre-calculation, fits well with the needs of an interactive authoring tool, where the user can not only sketch and extend 2D vector textures but also combine them seamlessly.

**Index Terms:** Computing methodologies—Computer Graphics—Graphics systems and interface—Texturing.

## 1 INTRODUCTION

From fibers and cellular organisms at microscopic scales to sea-weeds, schools of fishes, human queues, and alignments of trees or buildings at a larger scale, anisotropic distributions of shapes are ubiquitous in nature and man-made environments. Moreover, such structured shape-patterns have been used extensively in 2D for decorative purposes, from mosaics and wallpapers to the distribution of windows and architectural decorations on building facades. The perceived structure emerges from the anisotropy of these shape distributions. In particular, the specific ranges and variances of perceived orientations, both in terms of salient shapes and alignments, convey their unique visual appearance. This work explores the synthesis of such structured 2D shape-patterns from a sketch.

Example-based texture synthesis has already been extensively studied. However, existing methods have mostly focused on point distributions. They have achieved statistical accuracy using noise models, continuous representations of discrete distributions such as pair-correlation or probability-density functions, or neighborhood metrics and energy optimization. The few methods that address anisotropic distributions of shapes have used multiple point samples or proxy geometries to perform structured pattern analysis and synthesis. To the best of our knowledge, none of them have addressed the case of anisotropic shape-patterns that can include both bounded and unbounded (fiber-like) shapes. While the use of deep learning may be a promising alternative, it requires large training databases

---
*e-mail: paulinehnolivier@gmail.com

†e-mail:memari@lix.polytechnique.fr

‡e-mail:marie-paule.cani@polytechnique.edu

and long precomputation (learning) times, which has limited its use, so far, in interactive design scenarios.

This work tackles the interactive, sketch-based design of anisotropic distributions of shapes in 2D. Given any sketched pattern (a distribution of simple bounded shapes and/or fiber-like unbounded shapes), our method efficiently synthesizes a perceptually similar, consistent, and non-repetitive distribution of shapes in an extended 2D domain. Note that the input pattern is fully preserved at the synthesis stage. Indeed, contrary to previous methods, the input becomes the central part of the extended texture while being seamlessly integrated into its larger surrounding. Our solution increases user control and also seems to improve the perceived similarity of the results. One such result generated by our interactive system is shown in Fig. 1 with its interface.

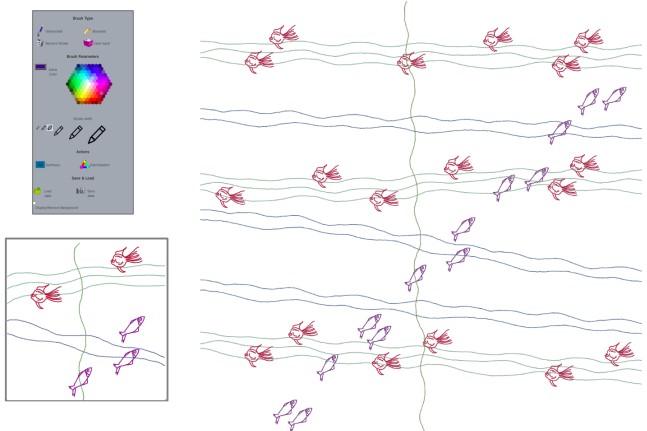

Figure 1: Based on a few perceptual and depiction hypotheses, our method extends an input sketch (bottom left) into a larger vector texture (right). Both bounded (individual fishes) and unbounded shapes (wavy lines) are seamlessly handled. The simple interface (top left) is quick to learn and easy to use.

Real-time analysis and synthesis of distributions require an efficient representation, encoding both local and global correlations between shapes. Our first insight is to introduce a compact encoding for anisotropic distributions, called the *Support Structure Hierarchy*, where individual supporting structures are lead directions of alignments or line skeletons computed from user strokes, all computed at various scales. This representation leads to a particularly simple and efficient multi-scale analysis of the distributions of orientations in the input sketch. It also enables efficient domain extension.

The main challenge at the synthesis stage remains to understand user expectations and the required criteria for perceptual similarity. The (new) case of fiber-like shapes is particularly challenging because extending fibers that are disjoint in the input exemplar may generate intersections in the extended domain. This could strongly affect our perception of the output as looking different from the input. To support our insights, we formulate a set of perceptual hypotheses to drive our synthesis solution; they were then validated through a user study. In particular, our solution interprets non-intersecting

fiber-like strokes as curves that could slightly bend to prevent intersection in the extended domain.

Thanks to its efficiency, we integrated our solution into an interactive authoring tool, where users can progressively test and refine their designs. They can generate a wider variety of vector textures by interactively hybridizing features extracted from several input exemplars, e.g., combining shapes from an exemplar with larger-scale alignments from another exemplar.

In summary, the contributions of our work are threefold, as we introduce:

- a fine-to-coarse analysis method that hierarchically clusters user strokes into a Support Structure Hierarchy, based on a new "perceived distance" between line segments within a domain, depending on both their position and orientation;

- a coarse-to-fine synthesis method that extends the pattern around the input sample based on the extracted hierarchy and a set of perceptual hypotheses validated by a user study;

- an interactive authoring tool, enabling both domain extension and hybridization of structured shape-patterns.

## 2 RELATED WORK

This work addresses the 2D sketch-based synthesis of anisotropic discrete distributions. It is related to example-based synthesis that aims at generating an output that minimizes some statistical or perceptual distance from the input while avoiding artifacts such as salient repetitiveness. We focus below on distributions of 2D shapes, i.e., vector textures formed by arrangements of discrete 2D shapes, and also discuss recent alternatives based on deep learning. We refer the reader to the surveys by Wei et al. [21] and Gieseke et al. [7] for a more general overview.

**Discrete vector textures (or shape-patterns)** were generated by analyzing the distributions of individual shape centroids, then applying a two-step synthesis for the new centroids, followed by retrieving the associated shapes. The pioneering work of Barla et al. [3] aims at synthesizing stroke patterns. Their method computes a Delaunay triangulation from the centroids to retrieve the input distribution connectivity. During synthesis, they rely on a Lloyd relaxation and some perturbation to generate a new set of points from which the shapes are recovered. In the same mindset, Ijiri et al. [9] explore local growth processes before the relaxation process. However, these two methods are limited respectively to quasi-uniform distributions or 1-ring neighborhoods. To handle more general shape distributions, Hurtut et al. [8] define the input distribution as a combination of Gibbs point processes from which they generate a new arrangement using Monte-Carlo chains. However, all these methods, as well as the patch-based method of AlMeraj et al. [1], are unable to analyze and synthesize structured inputs. In particular, they cannot handle anisotropic distributions of shapes such as elongated ones nor analyze correlations between shapes, orientations, and spatial alignments.

Instead of using a single centroid point, Ma et al. [15] characterize each input shape by several sample points. They rely on a neighborhood metric and an energy optimization process to insert individual shapes in a predefined output domain. Their approach has been extended to dynamic textures [14], stroke auto-completion [22], and adapted to other texture workflows [4,10]. While the use of multiple sample points has also been applied to the distribution synthesis of arbitrary shapes, these methods only address bounded shapes—as opposed to unbounded shapes—and require post-processing to avoid inter-penetrations at the synthesis stage, which precludes their use in real-time.

Rather than sampling the input shapes, Landes et al. [11] propose to simplify the input shapes into proxy geometries. They introduce a spatial relationship measure that takes into account space

between pairs of shapes and their relative orientations. By extending the stochastic models to point distributions [16,23], their synthesis method successfully maintains distributions of distances and relative orientations of shapes. Although it handles anisotropic distributions, their method does not offer real-time performance and is limited to distributions of bounded shapes. However, we follow their proxy geometry's idea when we first reduce our bounded shapes into support segments (or central points) and bounding boxes.

In contrast, Roveri et al. [18] present the first example-based distribution synthesis method applicable to both bounded and unbounded shapes. Regardless of their dimension, shapes are decomposed into point samples that are encoded in a functional representation. A similarity measure is defined in the associated functional space to quantify the similarity between input and output. The synthesis is achieved in a few minutes through neighborhood matching and energy optimization. Like most other neighborhood-based texture synthesis methods, their method requires input patterns with enough repetitions to avoid bad local minima in the optimization, which would distort the synthesized structures. Moreover, contrary to our method, the use of a fixed neighborhood size prevents their method from capturing repetitive structures at different scales.

**Deep learning methods** have recently been applied to texture synthesis [6, 12, 17, 19, 20, 24]. They show promising results for capturing, at least partially, local and global correlations present in an input exemplar. In particular, Fish et al. [6] allow for sketch stylization via the transfer of geometric textural details from different images; this is related to our secondary goal of pattern hybridization. However, most of these frameworks are image-based and do not extend well to the discrete distributions of vector shapes, which is the scope of our work.

The method of Tu et al. [20] is closest to our goal of handling vector shape distributions; it characterizes point patterns via a trained VGG network. Our method, based on a simpler but efficient analysis stage, has the advantage of requiring tedious precomputation (which is inherent to deep-learning techniques) while achieving real-time processing of any newly-created input.

## 3 OVERVIEW

### 3.1 Hypotheses on Depiction & Perception

Extending a sketched pattern in a perceptually similar way requires making some hypotheses about the user's depiction and perception of the resulting pattern. Our key assumption, common to most sketch-based modeling systems, is that users see their input as a *general view* of the distribution [1] they want to create. Therefore, we expect the input to include all the necessary information in a perceptually representative way. This led us to three design hypotheses:

$H1$: **Groupings and alignments are meaningful:** All alignments and groupings are intentional.

$H2$: **Repetitiveness is explicit:** All the shapes that a user wants to see repeated in the output, are repeated in the input.

$H3$: **Non-overlapping shapes should remain disjoint:** Shapes that do not overlap the input should not overlap in the output.

These three hypotheses are used as guidelines for our method at the design stage, and then validated by a user study (see Sect. 6).

---

[1] In this work we call "distribution of strokes" any arrangement (or collection) of strokes having constant, or approximately constant, statistical properties over their spatial domain.

## 3.2 Creation and Preprocessing of an Input Sketch

During a sketching session, the user successively draws strokes of any color in a square representing our 2D Input Space (*IS*). See Fig. 1, left. We provide two different pens to denote bounded and unbounded strokes. The first ones are limited to the dimensions of *IS*, while the second ones are interpreted as extending beyond the input domain, either in both directions if both extremities reach the border of the *IS*, or in a single direction (in case an unbounded stroke does not reach any border of *IS*, we add a segment to connect it to the closest border). The data stored for each stroke are a list of points, a color, a thickness, a type (bounded or unbounded), and a principal direction computed on the fly from the Principal Component Analysis (PCA) on the coordinates for all points of a stroke.

The user may sketch the input pattern in any order. As several stokes can be used to represent a shape, we provide an automatic clustering mechanism, presented next, to identify shapes at the beginning of the analysis stage.

## 3.3 Processing Pipeline

Multi-scale analysis: In the same spirit as StrokeAggregator [13], we rely on our perceptual hypotheses to reduce the input strokes into coarser structures. In particular, our analysis stage consists in iteratively extracting a fine-to-coarse hierarchy of support structures (the Support Structure Hierarchy) from the input strokes according to alignments and multi-scale repetitions in the input (see Fig. 2). We first cluster bounded strokes into *shapes* composed of one to several strokes, and we consider each unbounded stoke as an individual shape (**Level 0**). Note that colors are not used in the clustering, enabling the use of several different colors in a given shape. Bounded shapes are then simplified either into a *central point* or a *support segment* depending on their degree of anisotropy (Fig. 2c). Central points and support segments are clustered according to both orientation and position to find alignments and then grouped into *fibers* (Fig. 2d), forming the **Level 1** of the Support Structure Hierarchy. Other fibers are directly extracted from the unbounded strokes (Fig. 2c'). To capture repetitions at a larger scale, fibers of similar orientation are clustered into *fiber medians* (**Level 2**, Fig. 2f), which are finally grouped into *lead directions* (**Level 3**, Fig. 2g). During this hierarchical clustering and simplification process, we progressively partition the input domain *IS* into a hierarchy of *ribbons* that express the variability of position of each substructure around its parent structure. We use this partitioning to allow an adequate degree of variability while avoiding unwanted overlaps at the synthesis stage. See Section 4 for details.

Synthesis stage: Unlike most existing approaches, our method to synthesize distributions consists in directly replicating local and global correlations between the input shapes, encoded by our Support Structure Hierarchy. To avoid exact repetitions, this is done by instantiating each structure from top to bottom of the hierarchy while perturbing their positions within adequate *allowed areas*. We compute these areas to prevent overlaps between strokes belonging to the same lead direction and at a low cost since no further overlaps detection will be required.

The structures at the top of the hierarchy are first extended to the user-selected larger 2D domain, defined as a radial extension of ratio $k > 1$ of *IS*. The Support Hierarchy is then traversed from top to bottom and down to the individual strokes. At each level, the repetitive structures are repeated within the larger domain to generate the extended structured pattern. This is done following our design guidelines: a shape that appears only once in the input (such as the vertical seaweed in Fig. 1) will be extended at its extremities in case of an unbounded stroke, but will not be repeated (consistency with $H2$). In addition, at each level of the hierarchy, the allowed areas within ribbons are used to guide the synthesis of substructures while preventing unwanted overlaps (consistency with $H3$). Note

that curving some of the supporting structures is necessary to avoid undesired overlaps in the extended domain, as illustrated by the three green waves that do not overlap with the two blue waves in Fig. 1, right. This process, an original step of our solution justified by our perceptual guidelines, will be detailed in Section 5.

Interactivity and hybridization With its real-time performance, our method not only allows users to sketch and extend a given shape-pattern, but also to return to the sketching interface to iteratively improve their input. In our authoring system, all identified shapes are recorded in a shapes database, allowing the user to refine the input by adjusting their position or to reuse them later for another design. The hierarchical structures extracted from the analysis stages of different inputs can also be combined to create a different design, a process called *hybridization* (see Section 6).

## 4 FINE-TO-COARSE ANALYSIS

### 4.1 Level 0: From Strokes to Shapes

As illustrated in Fig. 2b,b', the bounded and unbounded strokes in the input are analyzed separately to extract supporting lines that will then be processed in a combined manner.

We consider the unbounded strokes as individual unbounded shapes. In contrast, we extract the bounded shapes by grouping the input bounded strokes as follows. We compute the oriented bounding box of each bounded stroke and group these boxes according to their pairwise distances. We then associate the resulting bounded shapes to a single central point or support segment, according to an anisotropy threshold. The resulting set of support segments and central points is the first simplification of the input, efficiently encoding the principal directions and approximate positions of the bounded shapes.

### 4.2 Level 1: From Shapes to Fibers and their Ribbons

We approximate each unbounded shape with a line, called fiber, that best matches its principal direction and position. This support line, augmented with a perpendicular thickness to cover the whole shape, is called a ribbon. For bounded shapes, finding such fibers and ribbons requires analyzing anisotropic information such as alignments. We retrieve the support lines of support segment and cluster them using the Mean Shift algorithm. We then compute a central fiber within each cluster. The central points are first clustered by position, before using Principal Component Analysis to compute their main directions of alignment. Representative fibers are defined from the centroid of each cluster and these principal directions. Thicknesses are computed for each of these fibers, so that the corresponding ribbon fully covers the shapes associated with the clustered points or segments.

### 4.3 Level 2: From Fibers to Fiber Medians

At this stage, we group the fibers with similar orientations and close positions. Since we focus on anisotropic distributions, we prioritize the orientations of fibers over their positions. We first compute the histogram of fiber orientations to group those belonging to the same anisotropic distribution. We then refine each cluster using a specific *perceived distance*, which we define as the minimum distance between each fiber intersection points with the domain contour (see Fig. 3). This distance takes into account both the position and the orientation of the lines: the more parallel and closer two lines are to each other in position, the smaller the distance. We store each resulting sub-cluster as a fiber median defined as the mean of parameters both in orientation and in position of the clustered fibers (see Fig. 2f). We also record the circular standard deviation associated to each fiber median for later use at the synthesis stage.

Similar to the previous hierarchy level, a thickness parameter is associated with each newly created fiber median to define an associated ribbon that fully includes the sub-ribbons of the clustered substructures (see Fig. 4).

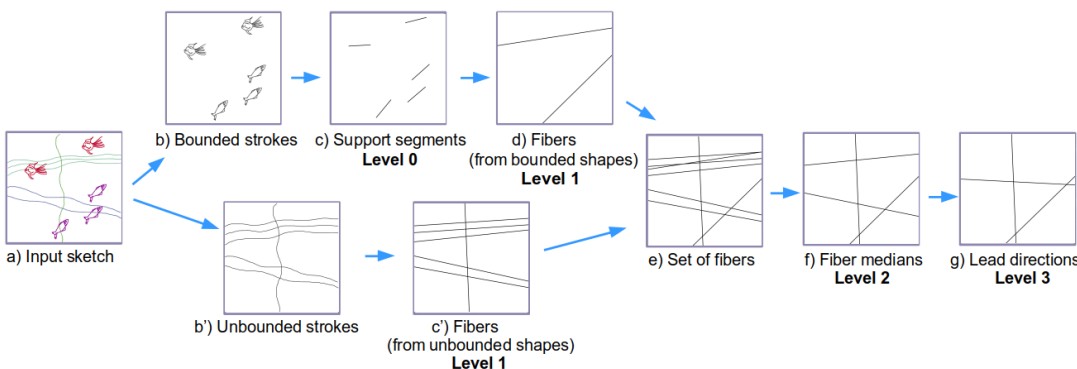

Figure 2: Processing pipeline for the fine-to-coarse analysis of a sketch into a Support Structure Hierarchy.

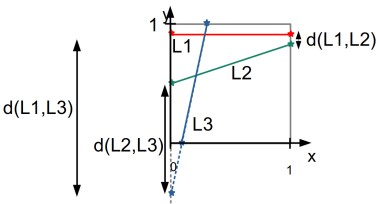

Figure 3: We compute the "perceived distance" between two fibers in a normalized input domain. It is defined as the minimal distance between their intersection points on any of the lines bordering the domain ($X = 0, X = 1, Y = 0, Y = 1$), which is extremely fast to compute (for each fiber, only the 4 values $y_{X=0}, y_{X=1}, x_{Y=0}, x_{Y=1}$ are needed). This distance accounts for both position and orientation, and is defined even if the lines intersect in the domain. Here, $d(L1, L2) < d(L2, L3)$, which matches our perception.

### 4.4 Level 3: From Fiber Medians to Lead Directions

The top-level of aggregation in our hierarchical analysis aims to group similarly oriented fiber medians. We use the same clustering process as the previous level of the hierarchy. We represent each cluster by a lead direction, defined using the average of the clustered elements in orientation and position (see Fig. 2g). As at the previous hierarchy level, ribbons are defined by associating a thickness parameter to each lead direction to include ribbons around clustered median fibers. As a result, the input space *IS* is divided into nested ribbon-like structures (see Fig. 4).

### 4.5 Computing the allowed displacement areas

The last step of the analysis stage is to calculate the available space around each clustered shape or ribbon within their parent structure in the hierarchy. We call this space the *allowed displacement area*, as it will be used at the synthesis stage to add random displacements to repeated structures, providing visual diversity while avoiding unwanted overlap between shapes.

Displacement areas for ribbons   Starting at the top of the hierarchy, we recursively decompose each ribbon, using splitting lines parallel to its main axis, and evenly divide the empty space between neighboring non-overlapping sub-ribbons (defined as ribbons around one of the clustered substructures). The distance between neighboring sub-ribbons (i.e., the minimum distance between their contents) used to position these lines is computed while considering a toroidal topology for *IS*. Based on this distance, two lines are evenly generated between the neighbouring ribbons to define the limits of extended regions for each of them, as well as an empty space between them.

This decomposition results in a displacement region around each sub-ribbon, and a given distance, called *gap* between them. Note that since we use the parent orientation for this decomposition, the sub-ribbons usually have a slightly different orientation. Moreover, they are not necessarily centered in the associated displacement region (see Fig. 4) a) and b).

Finally, the minimum and maximum values of the gaps are stored in the parent structure, together with the set of displacement areas associated with sub-ribbons.

Displacement areas for bounded shapes   These rectangular regions, delimited by dashed lines in Fig. 4 d), represent the areas within the fiber ribbon of a bounded shape in which its bounding box will be allowed to move during instantiation. Their two axes $(x, y)$ respectively correspond to the direction of the associated fiber median and its orthogonal direction. The allowed perturbation along $x$ (tangent to the direction) is set to half the distance to the next bounding box of a bounded shape, ensuring no overlap at the synthesis stage. The allowed perturbation along $y$ is set so that the bounding box can cover the whole associated fiber-median ribbon. Again, these computations are performed considering a toroidal topology for the input space *IS*. Therefore, the computed displacement areas can then expand outside *IS* (see the orange areas in Fig. 4 d), which is less restrictive when extending the pattern to a larger input domain.

## 5 SYNTHESIS OF AN EXTENDED SHAPE-PATTERN

To enable seamless exploration of a larger 2D domain by simply zooming out after sketching, our objective is to retain the user-drawn strokes within *IS* while extending and repeating them in a larger output space *OS* (defined as an expansion of *IS* by a ratio $k > 1$). This is done by a coarse-to-fine process in which the elements stored in the Support Structure Hierarchy are extended to *OS* and repeated as necessary.

Extension and repetition of lead ribbons:   According to $H2$ (see Sect. 3.1), lead directions consisting of only one fiber median, (such as the vertical lead direction in Fig. 5) should not be repeated. Therefore, we simply extend them as well as their unbounded child structures to span the whole *OS*.

For the remaining lead directions (corresponding to the repeated sub-structures in the input), we perform the same extension to the entire *OS* but also generate new copies of the structure in the remaining space through an efficient randomized repetition procedure, as follows. For each lead ribbon, we start from a displacement area with a single neighbour, randomly generate a new gap using values in the recorded range, and generate the next displacement area as to randomly clone one of the existing ones (i.e., using the same width). We apply this technique to progressively fill *OS*. The randomness in the gap values between displacement areas for sub-ribbons generates

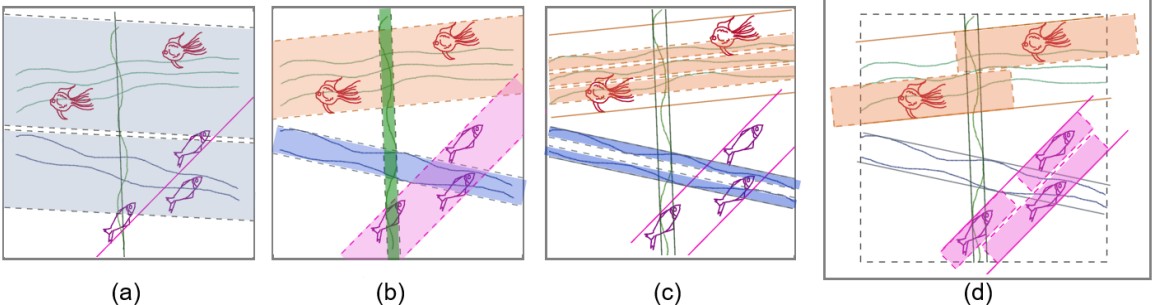

(a)         (b)         (c)         (d)

Figure 4: Input domain partitioning: (a) lead ribbons; (b) ribbons around the fiber medians; (c) the sub-ribbons inside the ribbons; (d) the displacement areas for bounded shapes.

different lead ribbon configurations, and thus different outputs from the same input (see Fig. 5).

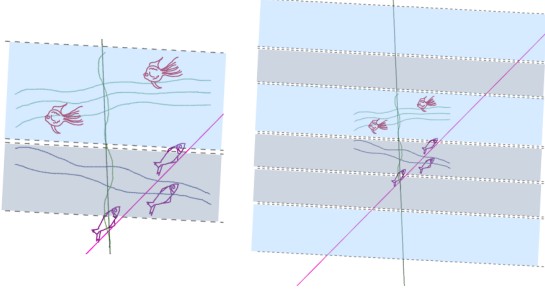

Figure 5: (left) Allowed displacement areas between dashed lines, based on lead directions; (right) Randomized repetition and propagation of lead ribbons.

Fiber medians ribbons will now be generated within the newly extended and repeated lead ribbons, as presented next, at the cost of slightly bending some of them as well as their child structures if they happen to overlap when extended to $OS$.

**Repetition of fiber medians and ribbons**  For each newly generated displacement area, we synthesize its fiber median by first copying the parameter values of the original ribbon. We then use the circular standard deviation on the medians' orientations computed during the analysis stage (Sect. 4.3) to perturb its orientation. We also perturb the position of its centroid to place it in the middle of the displacement area.

While the middle part of generated ribbons is guaranteed to remain within their lead ribbon, this is not necessarily the case when they extend to $OS$, as illustrated in Fig. 6 (left). When this occurs, we slightly bend a ribbon and its fiber median (see Fig. 6 (right)) to make it fit entirely inside its allowed displacement area.

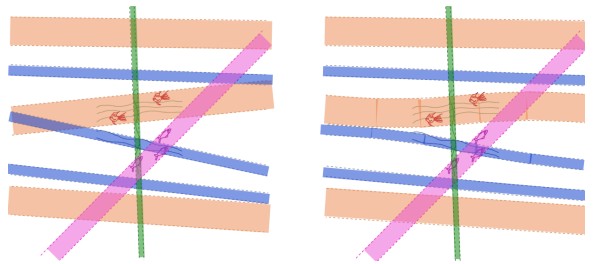

Figure 6: Ribbons repetition in $OS$: (left) without any bending; (right) with slight bending.

**Avoiding overlaps by bending structures**  Inspired by the physical properties of (real) fibers, we consider the following assumption: the thinner the ribbon, the more flexible it may be. This can be formalized through the equation $R = \tau w$, relating the curvature radius $R$ to the ribbon width $w$ and a stiffness parameter $\tau \in \mathbb{R}^+$.

In case of overlap, each ribbon will intersect twice with its lead ribbon. For symmetry reasons, we then bend both sides of the ribbon, even if one of the intersection regions is out of the output domain. To preserve continuity between the original borderlines of the ribbon and their curved version, we consider the midpoints ($M_1$ and $M_2$ in green in Fig. 7) between a projected point and the other intersection point as inflection points. For each of these inflection points (say $M$), the key idea is to find the arc of circle $C$ that passes through $M$ and remains inside the lead ribbon as illustrated in Fig. 7. We provide the details of this computation in the associated Supplementary material.

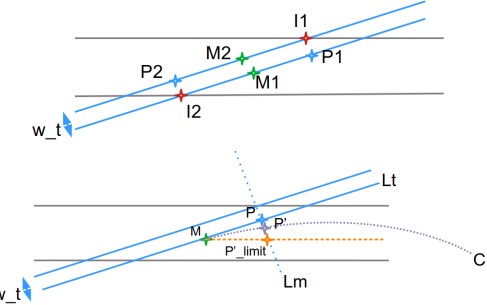

Figure 7: (Top) A ribbon has two intersections ($I1$ and $I2$) with its lead ribbon. (Bottom) The ribbon is bent to remain within its lead ribbon.

The same bending process is applied to the children's sub-ribbons to fit them inside their parent's curved ribbon.

**Shape distribution synthesis**  The final step is to synthesize new shapes within each extended or newly created fiber ribbon.

**a) Unbounded shapes**  We define four categories of unbounded strokes (lines, rays, arcs, and curves) that represent perfectly linear unbounded strokes, half-lines, unbounded strokes with a single curvature extremum in $IS$, and unbounded strokes with more than one curvature extrema, respectively. We begin by extending these unbounded shapes to $OS$ along their fiber direction, which is trivial for lines and rays. The arcs are extended through an alternative mirror duplication that leads to a smooth sinusoidal curve. The curves are first cut at their first and last extrema. Then, we alternatively mirror the version of the curve segment to extend it to $OS$ as illustrated in Fig. 8. These extended strokes are stored in the local frame of their corresponding fiber. They will therefore be

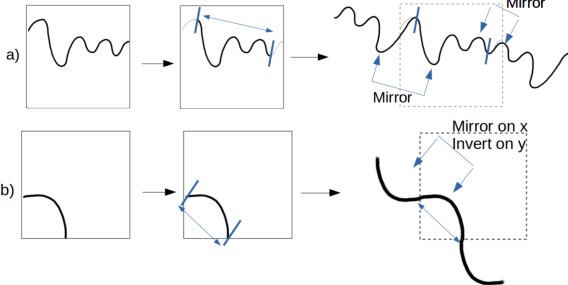

Figure 8: Extension of unbounded strokes: (a) curve; (b) arc.

automatically repeated and curved if necessary through the repetition process of their parent structures in the hierarchy. The resulting curved structures are shown for different sizes of *OS* in Fig. 9.

b) Bounded shapes    We process bounded shapes by first iteratively repeating their representative support segments or central points along their extended fiber while using the previously computed displacement areas to perturb their positions, and drawing the shapes in the resulting local frames. We then reuse their local positions relative to their fiber to repeat them within the parent fiber medians ribbons, but with randomly modified positions in the allowed displacement areas. See Figure Fig. 10.

Avoiding residual overlaps:    Given that repetitions in different lead directions are computed independently, lead ribbons with different orientations may naturally intersect. This may lead to perceptual artifacts if these lead directions both contain initially non-overlapping bounded shapes. Indeed some undesirable overlaps may occur in the output. We use an *AABB* tree to partition *OS* and efficiently detect overlaps between the displacement areas of bounded shapes. In a such case, we restrict the corresponding displacement areas. If this strategy fails (not enough space to insert a shape), we do not instantiate it (see Fig. 11 (b) for such a challenging example).

## 6    RESULTS AND DISCUSSION

### 6.1    Interactive authoring system

We implement our prototype system in WebGL. Creating and extending highly structured vector patterns is made easy by our method, as shown in Fig. 11. In addition to the main sketching and texture expansion interface, the user may store and reuse complex shapes, such as the two categories of fishes in Fig. 1. The use of our system for creating a complex sketch, inspired by biology, is illustrated in Fig. 12.

In addition, several input shape-patterns can be interactively combined to create a hybrid one, as follows. Thanks to the Support Structure Hierarchy, the user can select the desired level of hierarchy from two different input shape-patterns and combine them and create an hybrid result. We rely on the fact that our Support Structure Hierarchy encodes the input data into structures defined in the local frames of their upper structure ribbons, themselves characterized by their main direction and width. Therefore, consistent patterns can be generated while the input shapes, fibers, fiber-medians, or lead directions are changed. Such a hybridization is shown in Fig. 13 and its different steps displayed at the end of the accompanying video.

### 6.2    Comparison with previous work

We compared our results with both distribution-based and deep-learning-based methods for generating vector textures from examples. Since most classical methods are limited to distributions of bounded shapes, we restricted comparison to this sub-case (see Fig. 14).

Since our results seemed close to those of the best classical method, Landes et al. [11], we selected this method for further comparison in our user study (see below).

We also tried our method on examples presented as failure cases in previous papers, such as Fig. 11 (a failure case of [5]) and Fig. 15 (a failure case of [4]). In both cases, our solution was robust and managed to maintain the regularity of the structured input for both bounded and unbounded shapes.

Lastly, we compared our results with those of Tu et al. [20], the only deep-learning method tackling point distributions (see Fig. 16). Although our method is interactive and does not require any precomputation stage (in contrast to the hours of training of deep learning methods), the quality of our results looks comparable to the first example while the method from Tu et al. [20] gets more of them in the second one.

### 6.3    User study

We carefully designed an online user study to validate the perceptual hypotheses presented in Sect. 3.1, as well as the perceived quality of the extended textures we generate (See our supplemental document for screenshots and detailed results).

This study was composed of two parts: an interactive drawing session and a comparison session. We let the reader refer to the supplementary material for an illustration of all the proposed tasks. During the drawing session, the users were asked to manually draw an extended texture from a given input pattern. For this interactive session, we sorted the tasks in increasing orders of complexity: the first task consists in replicating (H1) the bounded shapes and avoiding overlaps (H3); the second task was about respecting the groupings and alignments (H1) of unbounded shapes while also avoiding overlaps (H3); the third task was mostly about our repetitiveness hypothesis (H2) for unbounded shapes, and the last task was about the respect of groupings and alignments (H1) of bounded shapes while avoiding overlaps (H3). On the other hand, in the comparison session, the users were asked to select the closest result from a given 2D input. The first selection task was to check whether users preferred an output with overlapping unbounded strokes or not (H3). The objective of the second task was to verify our repetitiveness hypothesis (H2) for unbounded strokes. Each experiment lasted around ten minutes, most of which was during the drawing session.

The study was conducted by 35 users, from 19 to 61 years old, including 22 males, 9 females, and 4 genders unspecified. 14 had an intermediary or expert experience in digital design and 9 as traditional designers.

Among the guidelines to validate, $H1$ (Groupings and alignments are meaningful) was validated by the drawing session, where 97% of the users preserved the grouping of fiber-like shapes and 76% of the users respected the anisotropy directions of bounded shapes in their drawings. $H2$ (repetitiveness is explicit) was validated by most users during the comparison session and was also observed in the user's drawings as those of Fig. 17. $H3$ (non-overlapping shapes should remain disjoint) was validated as well by the users' drawings, with 73% of overlapping-free drawings when it was the case in the input.

As in the study of AlMeraj et al. [2], we rely on our user study to let users choose between our extended textures and the generated ones from Landes et al. [11] (shown in random order and using the same shape depiction) for the ants and the balloons examples of Fig. 14. Respectively 86% and 77% of users preferred our results. We attribute these unexpectedly good results to the fact we keep the exact input pattern at the center of the generated texture while seamlessly extending it sideways.

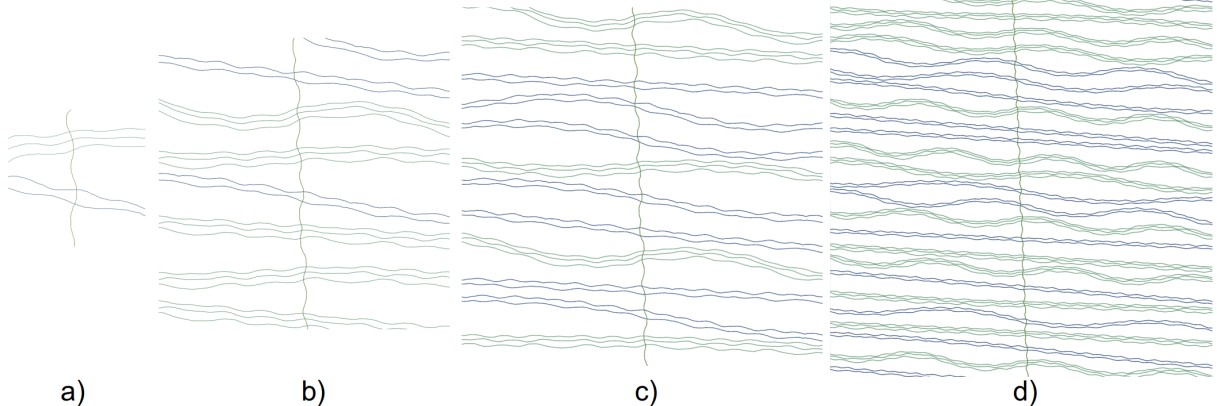

a)        b)        c)        d)

Figure 9: Variation of lateral ratio ($k$) for unbounded stroke distribution: (a) input; (b) $k = 3$; (c) $k = 5$; (d) $k = 10$. Note that these results have been scaled to fit in the figure.

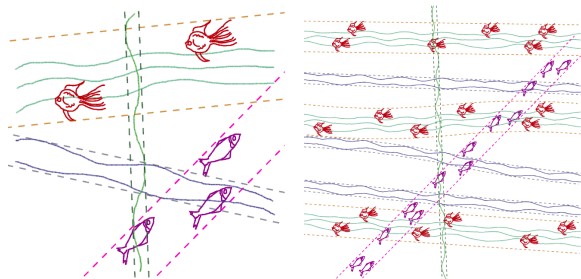

Figure 10: Synthesis outline: (left) input with ribbons between pairs of dashed lines; (right) shape repetition within the extended and synthesized ribbons.

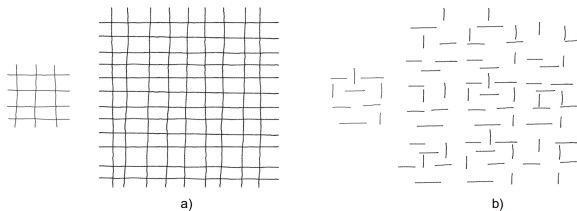

Figure 11: Our synthesis method maintains the perceived regularity of structured distributions (known to be hard to handle) in both cases of unbounded and bounded strokes.

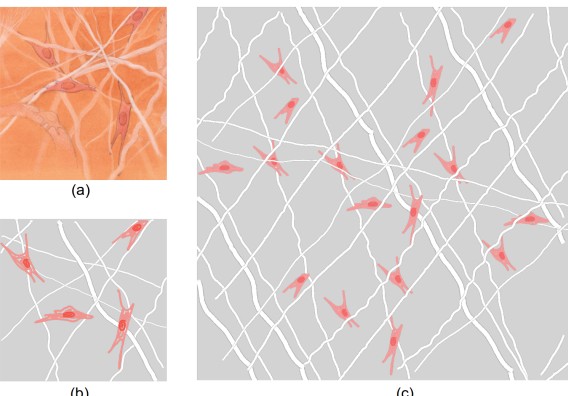

Figure 12: a) Biological illustration depicting cells that navigate in a distribution of fibers; b) Input sketch inspired from (a); c) Result.

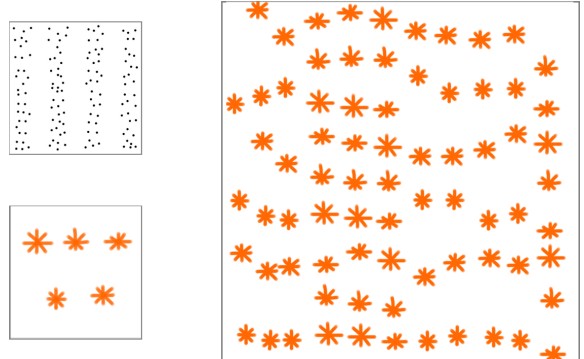

Figure 13: Hybridization example, where two input shape-patterns (left) are combined to create a new result (right). In particular, we combined the fiber medians from the top input shape-patterns with the shapes from the second input shape-patterns.

## 6.4 Performance

The following table was computed using the Google Chrome runtime performance on an Intel(R) Core(TM) i7-7920HQ CPU at 3.10GHz. The second column gives the number of points in the input example and then the time in milliseconds of respectively the analysis and the synthesis. Note that the synthesis has been performed with a ratio of $k = 3$. As can be observed, the overall computation times take less than a second.

| Example | # Points | Analysis | Synthesis |
|---|---|---|---|
| Fish & seaweeds (Fig. 1) | 7699 | 73ms | 111ms |
| Biology (Fig. 12) | 3094 | 27ms | 86ms |
| Ants (Fig. 14 top) | 9447 | 134ms | 233ms |
| Balloons (Fig. 14 bottom) | 4034 | 42ms | 68ms |
| Trunks (Fig. 15) | 3164 | 68ms | 83ms |

## 6.5 Discussion and limitations

The specificity of our method compared to previous work is that it does not require any neighborhood matching at the synthesis stage, given that our hierarchical representation already captures correlations. This leads to real-time performance suitable to our

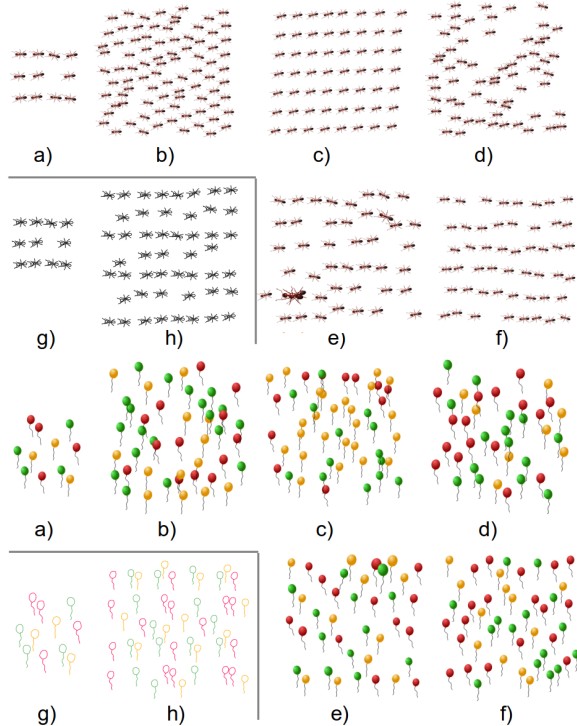

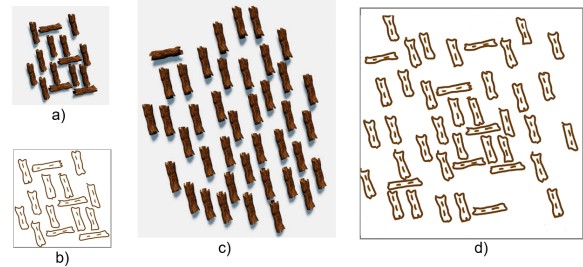

Figure 14: Comparison with distribution-based methods: (a) image input; (b)Barla et al. [3]; (c)Ijiri et al. [9]; (d)Hurtut et al. [8]; (e)Ma et al. [15]; (f)Landes et al. [11]; (g) our corresponding sketched input; (h) our result.

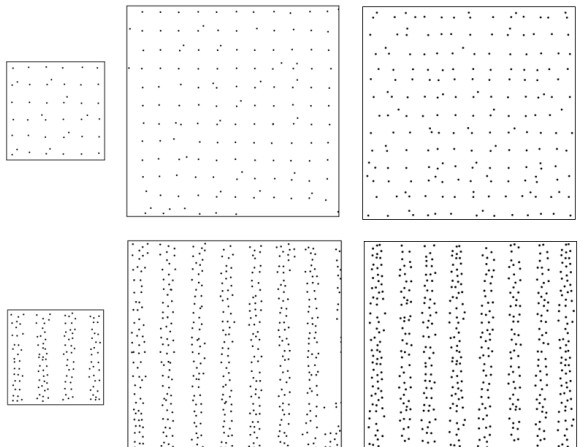

Figure 16: Comparison with the closest deep learning method [20]: (left) input distribution; (middle) results from Tu et al. [20]; (right) our results.

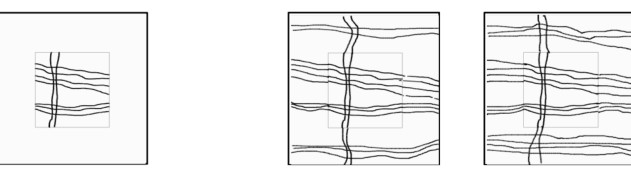

Figure 17: (Left) An example of input for the drawing session; (Right) Example of sketches created by different users

strokes is a design choice, which could also be disabled by the user if necessary.

Fig. 18 presents a failure case for our solutions, as bounding boxes around strokes overlap, while these strokes should not be grouped. To solve this problem, we could allow the choice between bounding-boxes-based distances and centroid-based distances as clustering criteria for bounded strokes. This would facilitate the processing of any dense distribution.

As the last limitation, extracting the main direction for unbounded strokes restrict their shapes to mostly linear ones. In particular, we do not consider branching curves or circular lines. In the case of branching, individual strokes would be split into isolated curves. This could result in unwanted overlaps during the synthesis. In addition, even if the grouping was forced, our current way of representing unbounded elements using a single linear ribbon of given width would fail. In the case of circular lines, we would need to cut them into linear pieces and treat each piece individually, which would result in handling branching. Therefore, seamlessly extending patterns that include such complex structures remains an open problem.

---

Figure 15: Challenging structured distributions: (a) input; (b) sketched representation of the input ; (c) result of [4]; (d) ours.

application context.

However, the notion of anisotropy, central in our method, makes it unsuitable to synthesize isotropic distributions. Indeed the computation of significant fiber directions becomes more difficult, which prevents further extraction of a structural hierarchy. Although this is the main limitation of our framework, in an authoring tool, our solution should be compared with a former method that handles isotropic distributions.

A useful extension would be to give the user the ability to choose among different perceptual hypotheses, for example, regarding explicit repetitiveness, which is not always desired or to allow overlaps between bounded and unbounded shapes. For example, in the biology illustration of Fig. 12, the cells depicted in pink/red should remain attached to the underlying fibers, which is not the case in our solution. Indeed, we never cluster bounded strokes with unbounded strokes, even if they overlap. This could easily be added as an option. Generating wavy curves rather than intersecting unbounded

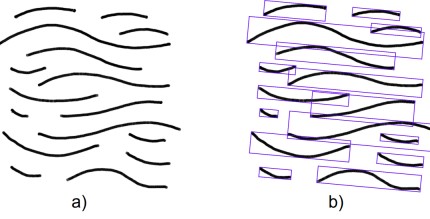

Figure 18: Input example from [15], where our current stroke clustering method fails.

# 7 CONCLUSION

Motivated by the interactive design of vector textures, we presented a multi-scale method to efficiently extract anisotropic properties from an input pattern, and seamlessly extend it to a larger 2D domain. Although our method runs in real-time, the visual quality of results is comparable to that of state-of-the-art vector texture generation methods, including those that require higher computational time and/or training data.

The new Support Structure Hierarchy we introduced is crucial for our method. Extracted at the analysis stage based on a new perceived distance between the salient anisotropic structures within the input domain, it allows us to efficiently capture and reproduce the multi-scale structures while maintaining a good level of visual diversity in the synthesized distribution of shapes. In terms of the interface, our system can be used to quickly design new vector textures by interactively creating new patterns or combining existing ones.

Future work   Although our solution is well suited for most structured shape-patterns, our use of linear ribbon-like shapes to capture multi-scale anisotropy prevents us from handling more complex and branched structures. Addressing this specific case would be an interesting avenue for future work. In addition, an open and challenging problem would be to generate a 3D texture from the 2D exemplar interactively sketched by the user. In cases such as biological illustrations, this would allow users to navigate in a 3D structure created from the sketch, leading to a better understanding of the depicted environment.

## ACKNOWLEDGEMENT

We would like to thank Pierre Poulin for his feedback that helped us improve this paper.

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
