# OpenReview forum: "Structured Shape-Patterns from a Sketch: A Multi-Scale Approach"
_graphicsinterface.org/Graphics_Interface/2022/Conference — GI 2022_

### Official Review · Reviewer_fwyR · 2022-04-07
**Very interesting topic, but I'm left with many doubts**

**Rating:** 5
**Confidence:** 4

**Review:**

This paper proposes an example-based geometric texture synthesis technique that is tailored for highly anisotropic exemplars, made up of (roughly) linear sequences of discrete elements, together with continuous elements.  The algorithm hierarchically groups elements into higher-level abstract representations of directional sequences, and uses that to drive a synthesis algorithm.  Continuous ribbons are capable of bending to avoid generating intersections that were not implied in the user-supplied exemplar.

The results are pretty good -- the technique seems to be doing at least *something* to conform to the three hypotheses articulated in Section 3.1.  I was definitely left wishing for more results, though.  In order to evaluate the success of an anisotropic method, I must discount some of the results presented in the user study, and the main results that remain (Figures 1, 12, and *maybe* 15 and 16) don't offer a very broad spectrum showing the algorithm's range.

My biggest criticism is that the writing is very choppy -- I found it difficult to make sense of the algorithm and to interpret the figures, which leaves me with a lot more uncertainty than I'd like in evaluating the manuscript.  Later, I'll offer some more detailed suggestions for improving the writing.  I note also that the 9-page paper is accompanied by a 6-page appendix.  Some of the details in the appendix feel essential for understanding or reproducing the algorithms; others feel important for understanding the user study.  This division feels like an attempt to make a long paper look like a short (or at least less long) paper, which rubs me the wrong way.

Here's a list of assorted questions and comments on parts of the paper:

 * I think these two papers might be relevant:

> AlMeraj et al., "Patch-Based Geometric Texture Synthesis"
> CAE '13: Proceedings of the Symposium on Computational Aesthetics

> AlMeraj et al., "Towards effective evaluation of geometric texture synthesis algorithms"
> NPAR '13: Proceedings of the Symposium on Non-Photorealistic Animation and Rendering

The first paper is yet another isotropic example-based texture synthesis technique, which might bear comparison against the other techniques in Section 6.2.  The second paper is relevant because they had professional designers perform synthesis by hand, as in Section 6.3.

 * When referring to previous work, there's a mild sense in which this manuscript claims to be an advance over techniques based on proxy geometry.  It might be better to downplay this somewhat -- to some extent, I'd say that support segments (Figure 2) and the bounding boxes used for clustering *are* a kind of proxy geometry.

 * So, the fish in the main result are clones of two distinct glyphs, right?  It's not totally clear from the paper, and in the video each one is drawn slowly over time, which is misleading if they're pasted in from stored clip art.

 * How does the clustering work?  If clustering is mainly by proximity (and not by shape similarity, for example), then were the fish carefully laid out so that the red fish would form one cluster, and the purple fish another?  If I had a horizontal row of red fish crossing a vertical row of blue fish, would that break the clustering?  That seems like something the artist has to be quite conscious of in designing their input.  (For what it's worth, I judge the upper-right red fish in the Figure 1 input to be closer to the topmost purple fish than to the other red fish, so I wonder if I'm missing something.)  How does all of this translate to the cells in Figure 12, which are all over the place?  Are they still clustered along a single fibre?

 * H2 says "repetitiveness is explicit".  Does that apply to clusters of finite objects, gathered on a fibre, or doesn't it?  I can't tell.  It's especially confusing, because there's a single path of purple fish in the Figure 1 output, but multiple paths of red fish.  Isn't that inconsistent?

 * It seems odd to me that the designer must use one pen for finite strokes and another pen for continuous strokes.  Isn't it possible to detect things like the long seaweed-like ribbons in Figure 1 automatically?  all the infinite ribbons in the paper look like things that could be detected trivially.

 * The synthesis of continuous geometry is repetitive by construction (rather than stochastic).  That's OK for small extensions of the exemplar, but Figure 9c is starting to look a bit repetitive, and Figure 9d is very repetitive.

 * Section 6.3 makes several claims like "97% of the users preserved the grouping of fiber-like shapes".  Was that measured quantitatively?  Judged by the authors?  I'd like to know how that was determined.

 * Also in Section 6.3: "We attribute this unexpectedly good results to the fact we keep the exact input pattern at the center of the generated texture, while seamlessly extending it sideways.".  That's testable, no?  Would it be possible to modify the an algorithm like Landes et al. to preserve the exemplar in the centre of the design?  And if that's where most of the judgment of quality comes from, maybe synthesis is easier than we make it out to be?

 * The comparison with previous work in Figure 14 isn't very convincing.  First, it doesn't tell us much about the main contribution of the paper: anisotropy.  Second, this paper's results look quite different from all the other results, making it difficult to know if user preference was due purely to the layout of objects, or other visual attributes.  Figure 6 of the supplement looks more like an apples-to-apples comparison, but... with a different technique that's not mentioned in the paper?  That's so confusing.  I also note that the dot weight of this paper's results in Figure 16 is quite different from the Tu et al. results, making a direct comparison quite difficult.

Here are additional lower-level comments on the presentation.

 * The whole manuscript could use a thorough proofreading pass.  There are spelling mistakes (e.g., "see-weed" -> "seaweed", "inflection" vs. "inflexion"), typos ("exiting" -> "existing") punctuation problems (e.g., "ie." -> "i.e.", extra hyphens, or hyphenated words that should just be glued together), lots of grammatical issues (subject-verb agreement, pluralization, etc.) and general messiness ("within in the authorized displacement areas. See Figure Fig. 10.").  The discussion section is especially poorly written, and the supplement needs a great deal of work as well.  BTW, say "less than a second" or "a fraction of a second"; don't say "less than a fraction of a second".

 * I'm all in favour of italicizing a newly defined term the first time it's used (e.g., "ribbon", "fibre median").  But please, please, *please* do so *only* the first time.  Don't italicize every single time: it makes the text so difficult to read!

 * Citations are parenthetical -- never treat them as nouns.  So, for example, instead of "We refer the reader to [5, 18] for more general surveys.", say something like "See the surveys by Gieseke et al. [5] and Wei et al. [18] for a more general overview."

 * A lot of the figures are complicated and difficult to parse.  It took me a long time to figure out that the "lead ribbon" is the *region* bounded by pairs of dashed lines in Figure 4a.  But then what's the magenta line?  What's the green vertical line?  I'm not sure I understand what's going on in Figure 13.  Figure 15's caption refers to (a), (b), etc., but the images aren't labeled.  Figure 14's caption is very hard to read, because it's mostly citations.  I'm left wishing for figures that more clearly illustrate the many moving parts in this algorithm.

In the end, I'm torn.  I like this topic a lot -- I'm happy to see new work on geometric texture synthesis, especially work that doesn't just fall back on the default of deep learning.  But this manuscript leaves me with too many doubts to champion it for acceptance.  I'd like to see this work published eventually, but I feel like there's too much to do to prepare it for immediate publication in GI.

---

### Official Review · Reviewer_jYRT · 2022-04-13
**This paper describes a method for tiling line drawings in a perceptually consistent but non-repetitive way. The technique identifies the underlying structures of a given drawing (created using a drawing tool with brushes that determine whether a shape should be bounded or not), repeats this underlying structure radially outward from the given tile, and then re-fills in the details.**

**Rating:** 7
**Confidence:** 3

**Review:**

These authors give a thorough analysis of their technique, including a user
study and comparison with other methods. The method is heuristic-based but
relies on three clear principles regarding how to interpret the user's
drawing. The user study (described in the supplemental materials) is unclear
about how images were annotated to support each hypothesis. The examples of
user drawings should include (and clearly label) both supporting and not-supporting examples for
each hypothesis.

---

### Official Review · Reviewer_3dJN · 2022-04-13
**Not much technical contribution, but otherwise good paper**

**Rating:** 6
**Confidence:** 4

**Review:**

The paper presents a new vector example-based texture synthesis method. Having two types of input strokes, bounded and unbounded, they group them, fit straight fibers/fiber medians with their 'ribbons'. Ribbons are then extended while bending them slightly to avoid introducing new overlaps. Within each ribbon, strokes are continued via a combination of mirroring and simple continuation. Bounded shapes are simply repeated.

It is a generally well-written and clear paper. While the contribution is not too enlightening (and even though the applications of the method are not terribly clear to me), perhaps more than the approach, I appreciate the hypotheses they formulate in 3.1 and some evaluation of those with the user studies -- that might be useful for the future work. In particular, I like the analysis and the handling of the unintentional intersections via small bends.

I was somewhat disappointed at the use of PCA for the unbounded strokes, because this assumes the stroke is mostly linear. The limitations section should be clearer about it (so far I see it talks about branching curves).

Also, there are too many details missing from the paper making it non-reproducible. Please add all the missing details (e.g. kernel/window size in the mean shift algorithm, # of bins in a histogram, etc.)

Other than that, I don't have major concerns, only the following minor ones:

	1. The word 'distribution' used in the paper for the collection/arrangement of curves has a precise technical meaning that is different from what the authors mean. Unless it's a standard terminology in the area (I haven't seen that), please consider replacing it.
	2. Related work, or maybe the description of the method should cite StrokeAggregator by Liu et al., due to the similarity of the clustering mechanism.
	3. Fig. 4: abcd take a lot of vertical space, please make it more compact
Fig. 4 needs better visualization of ribbons than the dashed lines, otherwise it's unclear dashed lines depict a boundary of a strip/ribbon. Transparent colored strips?

---

### Decision · Program_Chairs · 2022-04-17

**Decision:**

Accept

**Comment:**

Congratulations, your paper has been accepted to GI’2022!

Please take into account suggestions by the reviewers to improve the final version of the paper.  Specifically, some reviewers noted that additional details are required to reproduce the method, and some sections could be present more clearly.